# DOR3D-Net: Dense Ordinal Regression Network for 3D Hand Pose Estimation

Yamin Mao[1], Zhihua Liu[1], Weiming Li[1], SoonYong Cho[2], Qiang Wang[1] and Xiaoshuai Hao[1*]

[1]Samsung R&D Institute China–Beijing
[2]Multimedia System TU, SAIT, SEC, Korea

`yamin18.mao, zhihua.liu, weiming.li, soonyong.cho, qiang.w, xshuai.hao@samsung.com`

## Abstract

*Depth-based 3D hand pose estimation is an important but challenging research task in robotics and autonomous driving community. Recently, dense regression methods have attracted increasing attention in 3D hand pose estimation task, which provide a low computational burden and high accuracy regression way by densely regressing hand joint offset maps. However, large-scale regression offset values are often affected by noise and outliers, leading to a significant drop in accuracy. To tackle this, we re-formulate 3D hand pose estimation as a dense ordinal regression problem and propose a novel Dense Ordinal Regression 3D Pose Network (DOR3D-Net). Specifically, we first decompose offset value regression into sub-tasks of binary classifications with ordinal constraints. Then, each binary classifier can predict the probability of a binary spatial relationship relative to joint, which is easier to train and yield much lower level of noise. The estimated hand joint positions are inferred by aggregating the ordinal regression results at local positions with a weighted sum. Furthermore, both joint regression loss and ordinal regression loss are used to train our DOR3D-Net. Extensive experiments show that our design improves the SOTA methods by a large margin on popular benchmarks. The source code will be released.*

## 1. Introduction

As intelligent machines like robots and autonomous vehicles strive to be socially aware in human-centric environments, accurately perceiving human hands and their intentions becomes vital for robotics [8, 19] and autonomous driving [32, 34]. High-quality hand pose estimation is an important prerequisite for accurately perceiving and interpreting human hand movements. With the development of deep learning, hand pose estimation from RGB images [1, 14, 24, 35] and depth images [3, 21, 23, 30] have attracted much attention. This paper focuses on improving

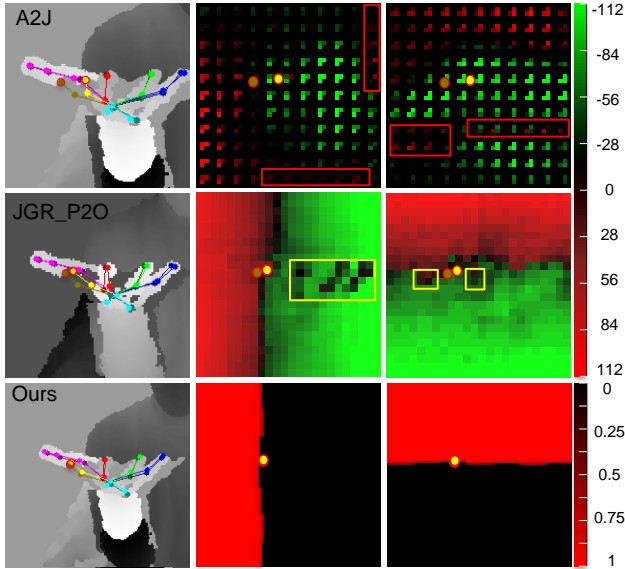

Figure 1. Visualization of final and intermediate results for comparison between SOTA methods and our method. Row 1 and 2 show predictions from A2J [30] and JGR-P2O [6] respectively and row 3 shows our predictions. For each row, column 1 shows the final result of prediction for an exemplar hand joint (superimposed on input depth image) and column 2 and 3 show the $x$-offset and $y$-offset maps respectively. For better comparison, bright and light yellow dots surrounded by red circles show the predict joint and ground truth respectively. Notice that several error areas present in the offset maps from A2J and JGR-P2O (highlighted with red and yellow boxes). In contrast, our probability map is clean. This is brought by our dense ordinal regression design and empowers our DOR3D-Net to surpass SOTA methods in public benchmarks.

depth-based 3D hand pose estimation, which aims to output joint coordinates in 3D space from an input depth image.

Existing hand pose estimation methods with dominant performance employ deep learning models of different structures. Some models [4, 33] extract deep representations and then directly regress the joint coordinates or other forms of hand model parameters. Differently, more recent models explore dense predictions. They usually use a dense grid and for each grid position predict an offset vector that

---

*Corresponding author.

points to a joint. The densely predicted offset vectors form an offset map and then are used to infer joint coordinates. For example, A2J [30] proposes dense anchors and aggregates the offsets of these anchors to estimate each joint in the image plane. JGR-P2O [6] regresses joints by weighted averaging over all pixels' offsets in both image plane and depth space. For these methods, depending on the distance between a grid position and its target joint, the offset value varies in a large interval, especially for high-resolution images. However, large-scale regression offset values are often affected by noise and outliers. These flaws are difficult to completely remove and will propagate to subsequent steps resulting in degradation in the estimated joint accuracy. In this paper, we explore ordinal constraints to improve dense prediction methods for hand pose estimation. Specifically, as a point traverses in space along the scanline, the spatial relationship between the point's position to a target joint should vary smoothly with strict ordinal constraints. A closely related previous work [17] is ordinal regression which converts a regression task into a series of binary classifies with ordinal constraints. The ordinal regression has been proven to be useful for several tasks such as age estimation [17] and depth estimation [7].

To our best knowledge, we are the first re-formulate 3D hand pose estimation as a dense ordinal regression problem and propose a novel Dense Ordinal Regression 3D Pose Network (DOR3D-Net). Specifically, the design of DOR3D-Net includes: (1) The problem of hand joint regression in 3D is decomposed into sub-tasks of binary classifications. Each binary classifier is associated to a grid in 3D with different interval distributions in image and depth dimensions. Each binary classifier predicts probability of a binary spatial relationship between the position and a joint point. (2) The ordinal regression results at different local positions are aggregated to infer joint positions with weighted sum. This allows us using a joint position loss together with the ordinal regression loss to supervise our DOR3D-Net. (3) Three branches of networks are used for each of the three dimensions respectively.

The experiments show that the binary classifiers in our design are easy to train and yield much lower level of noise. Fig. 1 visualizes the offset maps predicted by A2J [30], JGR-P2O [6] and our probability map in image plane respectively. The first row shows the offset maps of A2J. Anchor offset values are wrong in several local areas (highlighted by red boxes). The boundary of zero offset value appears as a curve, which severely deviates from its ideal form as a straight line. For JGR-P2O, the learned offset maps also include apparent errors (highlighted by yellow boxes). In contrast, the probability maps generated by DOR3D-Net are much cleaner and well approximate those of their ideal forms. The main contributions are summarized as follows:

- We are the first to re-formulate the 3D hand pose estima-

tion as a dense ordinal regression problem and propose a novel Dense Ordinal Regression 3D Pose Network.
- Specifically, we propose Ordinal Regression (OR) module to decompose offset regression into sub-tasks of binary classifications with less noises and outliers. Furthermore, both joint regression loss and ordinal regression loss are used to train DOR3D-Net in an end-to-end manner.
- DOR3D-Net is remarkably superior to SOTAs on existing methods, revealing the effectiveness of our approach.

## 2. Related Work

### 2.1. Depth Image Based Hand Pose Estimation

This paper focuses on the depth image-based 3D hand pose estimation task. According to summary of a large-scale public challenge HANDS2017 [31], state-of-the-art hand pose estimation methods can be roughly divided into two categories: regression-based methods and detection-based methods. Regression-based method directly regress hand joint parameters with extracted global feature representation. DenseRecurrent [4] uses PointNet network to extract features and iteratively refines the estimated hand pose with a point cloud representation. Detection-based methods generate dense pixel-wise estimations with heatmaps or offset vectors from local features. V2V-PoseNet [16] uses 3D CNN network to extract a feature-based volumetric representation and estimates volumetric heatmaps. DenseReg [29] decompose 3D hand pose as 3D heatmaps and 3D joint offsets and estimates these parameters by dense pixel-wise regression. Compared with heatmap-based method with relatively high computational burden, offset-based methods achieve a better trade-off between accuracy and efficiency and can be adapted in resource-constrained platforms. A2J [30] predicts per-joint pixel-wise offset through a dense set of anchor points on the input image. JGR-P2O [6] proposes a pixel-to-offset prediction network to address the trade-off between accuracy and efficiency for hand pose estimation. HandFoldingNet [3] inputs 3D hand point cloud and acquires the hand joint locations based on point-wise regression. SRN [20] regresses the joint position through multiple stacked network modules to capture spatial information. TriHorn-Net[23] computes two complementary attention maps of each joint and uses appearance-based data augmentation to improve the accuracy of hand pose estimation.

### 2.2. Ordinal Regression

The ordinal regression method maps direct regression into multiple binary classifiers and learns to predict ordinal labels. By preserving the natural order and supervising with multiple rank labels, the ordinal regression methods [7, 12, 13] have been proven to achieve much higher

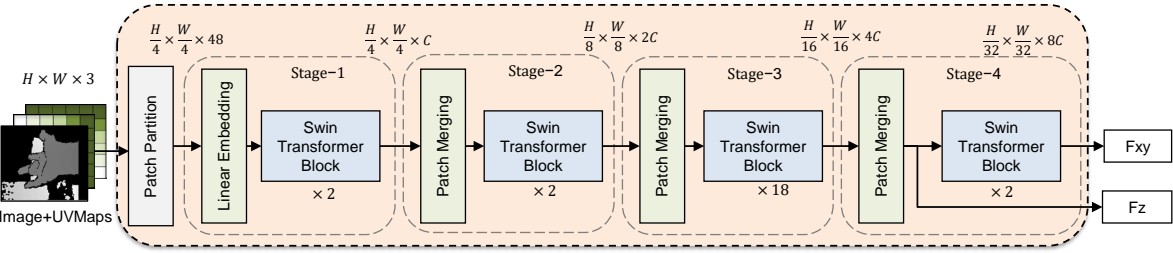

Figure 2. The pipeline of our transformer-based feature extractor. It contains patch partition and four Swin Transformer stages. Patch partition splits the image into multiple $4 \times 4$ patches and each patch is considered as a token. Then tokens pass through each stage to learn long-range feature interactions through Swin Transformer blocks. The final two feature maps from the last stage are sent into the dense ordinal regression module for 3D hand pose prediction.

accuracy and faster convergence than the direct regression. DORN [7] converts the depth prediction into an ordinal regression problem, which discretizes depth value into several intervals and obtains ordinal labels to improve depth estimation accuracy. Furthermore, [18] also proposes the definitions of ordinal depth, which are based on comparing the relative depth between different joints. However, [18] is different from our DOR3D-Net. In this paper, we reformulate the pose estimation problem into an ordinal regression problem and compare binary spatial relationships between a sampling position with respect to a joint point. Our proposed dense ordinal supervision guarantees the probability maps are ordinal, which reduces the depth noise effect and improves 3D joint pose estimation accuracy.

## 3. Methods

In this section, we first introduce the feature extractor module, which use a transformer-based feature extractor to learn dense local feature representations for capturing long-range relationships. Then, we elaborate on the details of dense ordinal regression module, which design to output pixel-wise probability maps and regress hand joints. Finally, we introduce the overall training procedure.

### 3.1. Feature Extractor

In the transformer-based feature extractor module (Fig. 2), the input image plane is split into multiple $4 \times 4$ patches with the patch partition module. Each patch is treated as a 'token'. Four Swin Transformer stages are used to learn attention among patch tokens for capturing long-range contextual information. These stages consist of linear embedding layers, patch merging layers, and Swin Transformer blocks with their structure details specified in [15].

Since the Swin Transformer structure contains only relative positional embedding, we modify the input by adding $U$, $V$ coordination maps (UVMap) and concat them together with the depth map to provide global absolute spatial information. $U$ and $V$ maps are generated by linear scaling

in Eq. 1, which corresponds to the in-plain coordinate map of each pixel.

$$U(i,j) = j/W, i \in [0, H), j \in [0, W),$$
$$V(i,j) = i/H, i \in [0, H), j \in [0, W). \tag{1}$$

Considering the in-plane xy regression and depth-plane z regression are quite different, following the design of A2J [30], two feature maps $F_{xy}$ and $F_z$ are output from 'Stage-4' with the same dimensions $\frac{H}{32} \times \frac{W}{32} \times 8C$. Then, both decoded feature maps are regressed with the dense ordinal regression module for 3D joint prediction.

### 3.2. Dense Ordinal Regression Module

Fig. 3 illustrates the pipeline for the dense ordinal regression module. With input as the learned feature maps, we utilize separate branches to estimate the three-dimensional coordinates of hand joints independently and output the predicted hand joint pose.

**Normal Discretization.** Since the hand joints reside in three dimensional space, we decouple the 3D solution space and quantize it by representative discrete values along each of the three dimensions. In image plane $x$-axis and $y$-axis directions, the intervals are $[0, W)$ and $[0, H)$ respectively. Uniform discretization (UD) is adopted to divide the image plane. Assuming that the intervals are discretized into $K_x, K_y$ sub-intervals along x-axis and y-axis directions respectively, the UD can be formulated as:

$$x_i = i * W/K_x, \quad y_j = j * H/K_y, \tag{2}$$

where $x_i$, $y_j$ are sampling points and then form well-ordered sets $S_x = x_0, ..., x_{K_x-1}$, $S_y = y_0, ..., y_{K_y-1}$. Here, we set $K_x = W/2, K_y = H/2$.

In the $z$-axis direction, we analyze statistics of the joints $z$ coordinate distribution and notice that it is close to normal distribution. Following this, the sampling interval of normal discretization (ND) becomes smaller as the sampling position becomes closer to the distribution center. In our specific

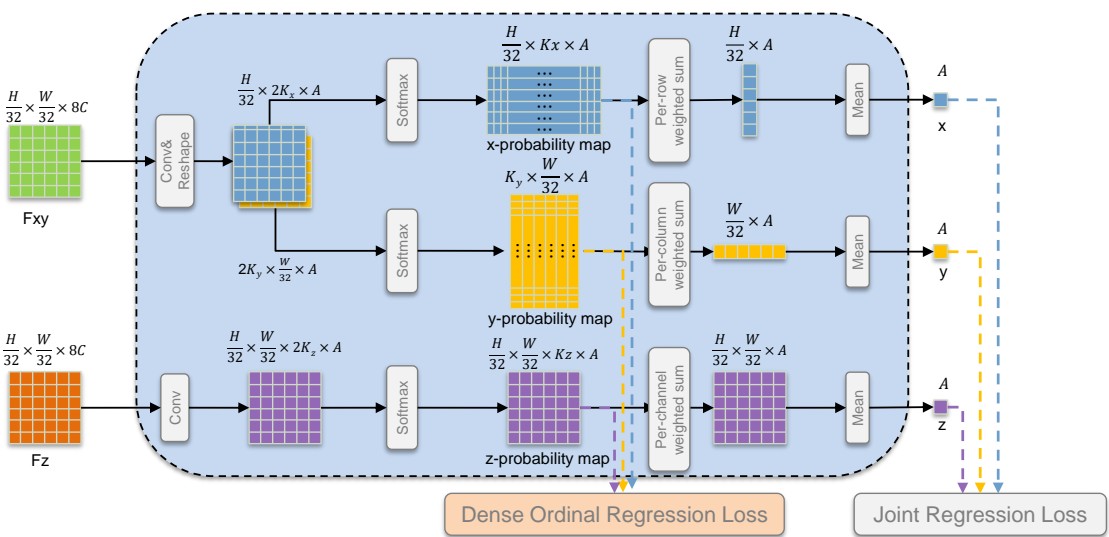

Figure 3. The pipeline of our proposed dense ordinal regression module. The inputs are two feature maps. With the reshape and softmax operators, we obtain binary probability maps. With weighted sum, the binary probabilities at local positions are aggregated to infer hand keypoints along each of the three dimensions respectively. Supervised by dense ordinal regression loss, these binary classifiers are easier to train and yield much lower level of noise, which helps to estimate accurate 3D hand joint poses.

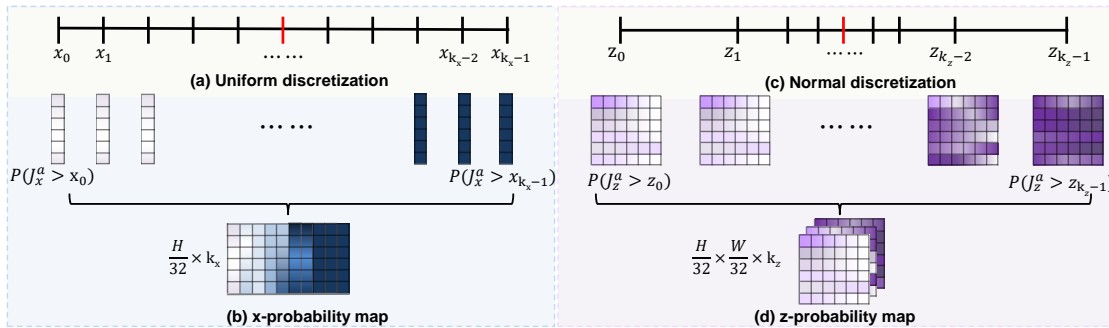

Figure 4. Visualization of the proposed $x$- and $z$-discretization process. $x$-axis uses uniform discretization and $z$-axis applies normal discretization. For the $x$-probability map, each column represents the probability that the keypoint is larger than the corresponding discretization threshold. For the $z$-probability map, each map represents the probability that the keypoint is larger than the corresponding discretization threshold.

implementation, we first divide the $z$-axis $[0, D)$ into several sub-intervals evenly, and then increase the frequency of sampling points by the exponential power of 2 for the consecutive sub-intervals to the midpoint $D/2$. Sampling points for the other half of the z-interval can be obtained by symmetry. These sampling points form a well-ordered set $S_z = z_0, ..., z_{K_z-1}$. Here, we set $K_z = D/4$, $D$ refers to the depth range of the cropped image. Fig. 4 (a) and (c) visualize the distributions along $x$ and $z$ respectively.

**Ordinal Regression.** After obtaining the discrete ordered classification sets $S_x, S_y, S_z$, we cast the hand pose estimation problem into an ordinal regress problem to learn the network. These well-ordered sampling points along x-axis, y-axis, and z-axis directions construct multiple bi-nary classification sub-problems. For each predicted coordinate of joint $a$, these binary classifiers are used to predict whether the hand joint location is larger than these discretization thresholds respectively and then form probability map $Prob_x$, $Prob_y$, and $Prob_z$. Here, coordination is represented as $J^a = (J_x^a, J_y^a, J_z^a)$.

$$Prob_x(i, j, a) = P(J_x^a > x_j), \, \forall \, i$$
$$Prob_y(i, j, a) = P(J_y^a > y_i), \, \forall \, j \qquad (3)$$
$$Prob_z(i, j, k, a) = P(J_z^a > z_k). \, \forall \, i, j$$

Fig. 3 illustrates the probability map generation process. Let $F_{xy} \in R^{\frac{H}{32} \times \frac{W}{32} \times 8C}$ denote the feature maps in the image plane, after convolution, reshape and softmax operators, the feature map $F_{xy}$ is converted into the probability

map $Prob_x \in R^{\frac{H}{32} \times K_x \times A}$ and $Prob_y \in R^{K_y \times \frac{W}{32} \times A}$. For the feature map $F_z \in R^{\frac{H}{32} \times \frac{W}{32} \times 8C}$, after convolution and softmax operators, it is mapped into the probability map $Prob_z \in R^{\frac{H}{32} \times \frac{W}{32} \times K_z \times A}$. $A$ is the number of joints. To guarantee the accuracy of classification, all these probability maps are densely supervised with the ground truth (GT) probability maps and introduced in section 3.3. Obviously, the ordinal regression solution only compares the binary spatial relationship between the keypoint and every discretization threshold. In comparison with dense offset regression methods, the solution space is reduced from a large interval to binary values which is easy to get the optimal solution and insensitive to the noise and outliers.

After getting the reliable probability maps $Prob_x$, we fuse the probability with its corresponding classification interval length and get the hand joint prediction vector in Eq. 4. The final predicted coordinate value $\hat{x}$ is the mean value of $\bar{x}(i)$. In the same way, we obtain the coordination $\hat{y}$ and $\hat{z}$ from $\bar{y}(j)$, $\bar{z}(i,j)$ in Eq. 4, respectively.

$$
\begin{aligned}
\bar{x}(i,a) &= \sum_j Prob_x(i,j,a) \cdot (x_{j+1} - x_j), \\
\bar{y}(j,a) &= \sum_i Prob_y(i,j,a) \cdot (y_{i+1} - y_i), \\
\bar{z}(i,j,a) &= \sum_k Prob_z(i,j,k,a) \cdot (z_{k+1} - z_k).
\end{aligned}
\tag{4}
$$

Here, $x_j$, $y_i$ and $z_k$ are sampling points. We concatenate the coordinations $\hat{x}$, $\hat{y}$ and $\hat{z}$ as the final results.

### 3.3. Loss Function

The network is jointly supervised by two loss: joint regression loss and dense ordinal regression loss. The binary probability maps along x-axis, y-axis and z-axis are supervised to guarantee that the learned features are robust to the low-quality depth image and have a reliable representation.

**Joint Regression Loss.** The GT hand pose supervises the neural network to generate an accurate hand pose localization at the final stage. The loss function is formulated as below Eq. 3.3. The endpoint error with smooth $L_1$ is used to compute the joint regression loss. Here, $L_{joint\_loss} = \sum_{a \in A} L_{smooth}(J^a - J^{a*})$,, where $J^a$ and $J^{a*}$ are the predicted coordinate and the GT of joint $a$, respectively.

**Dense Ordinal Regression (DOR) Loss.** To supervise the binary classifiers, we generate the GT of middle results (GT binary probability maps $Prob_x^{gt}$, $Prob_y^{gt}$ and $Prob_z^{gt}$):

$$
Prob_x^{gt}(i,j,a) = \begin{cases} 1, & if\ J_x^{a*} \geq x_j,\ \forall\, i \\ \\ 0, & otherwise.\ \forall\, i \end{cases}
$$

$$
Prob_y^{gt}(i,j,a) = \begin{cases} 1, & if\ J_y^{a*} \geq y_i,\ \forall\, j \\ \\ 0, & otherwise.\ \forall\, j \end{cases}
\tag{5}
$$

$$
Prob_z^{gt}(i,j,k,a) = \begin{cases} 1, & if\ J_z^{a*} \geq z_k,\ \forall\, i,j \\ \\ 0, & otherwise.\ \forall\, i,j \end{cases}
$$

where $J_x^{a*}$, $J_y^{a*}$ and $J_z^{a*}$ are the GT coordinates of joint $a$.

Take the $Prob_x \in R^{\frac{H}{32} \times W \times A}$ for an example, the DOR loss is defined as the cross-entropy loss to densely supervise all binary classification probability maps.

$$
L_{ord}\left(Prob_x, Prob_x^{gt}\right) = -\frac{1}{\frac{H}{32} \times A} \times
$$

$$
\sum (Prob_x^{gt} \log(Prob_x) + (1 - Prob_x^{gt}) \log(1 - Prob_x)).
\tag{6}
$$

The DOR losses in three dimensions are defined in Eq. 7.

$$
\begin{aligned}
L_{ord\_loss} &= L_{ord}\left(Prob_x, Prob_x^{gt}\right) \\
&+ L_{ord}\left(Prob_y, Prob_y^{gt}\right) + L_{ord}\left(Prob_z, Prob_z^{gt}\right).
\end{aligned}
\tag{7}
$$

**Total Loss Function.** The loss is defined as: $L = \lambda_1 L_{joint\_loss} + \lambda_2 L_{ord\_loss}$, where $\lambda_1$ and $\lambda_2$ are hyperparameters for balancing these terms. Here, $\lambda_1 = 3$ and $\lambda_2 = 2$.

## 4. Experiments

### 4.1. Experimental Setting

**Datasets.** There are four widely adopted datasets for 3D hand pose estimation task, including HANDS2017 [31], MSRA [26], ICVL [27], and NYU [28]. HANDS2017 dataset is composed of Big Hand 2.2M dataset [31] and the First-person Hand Action Dataset (FHAD) [9]. It contains 957K training and 295K testing depth images. MSRA dataset contains 9 subjects with each subject performs 17 hand gestures, and each hand gesture contains about 500 frames. ICVL dataset contains 22K training and 1.5k testing depth images with 3D annotations for 16 joints. The raw images with annotations are augmented to 330K samples by in-plane rotations. NYU dataset contains 72K training and 8.2K testing depth images labeled with 3D annotations for 36 joints. Following A2J [30], we only use 14 of the 36 joints from frontal view for both training and testing. Since the number of raw images in HANDS2017 dataset is over ten times bigger than any of the rest three datasets, we conduct ablation study on the HANDS2017 dataset.

**Implementation Details.** DOR3D-Net is trained with 2 NVIDIA V100 GPUs. We adopt the AdamW optimizer for all our experiments during training. In all experiments, the learning rate is $3.5 \times 10^{-4}$ with a weight decay of $10^{-4}$. The batch size for MSRA, ICVL and NYU datasets is 32, and the batch size for HANDS2017 is 64. For all datasets, the learning rate decays by 0.2 every 7 epoches. Similar to A2J [30] and V2V-PoseNet [16], we use hand center point to crop the hand region from an depth image and resize the image to $224 \times 224$. For MSRA, ICVL and Hands2017, the hand center points are released by V2V-PoseNet [16]. For NYU, we follow DenseRecurrent [4] and use the average location of the ground truth joints.

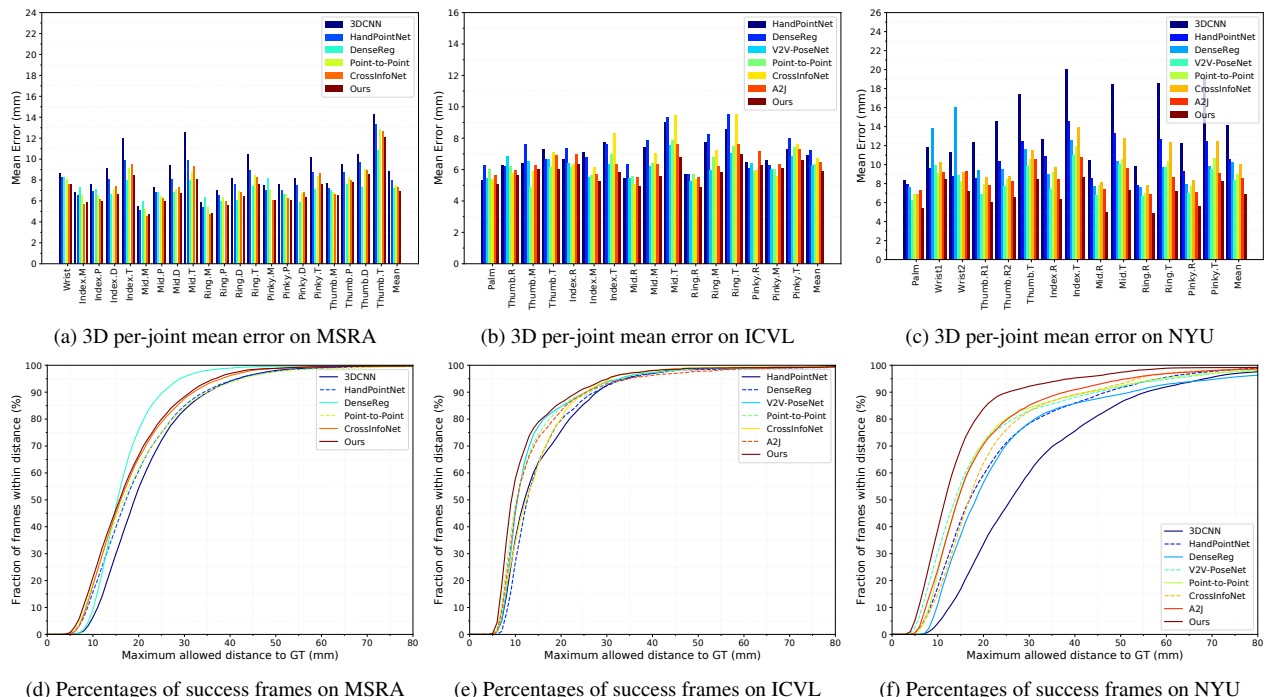

(a) 3D per-joint mean error on MSRA      (b) 3D per-joint mean error on ICVL      (c) 3D per-joint mean error on NYU

(d) Percentages of success frames on MSRA      (e) Percentages of success frames on ICVL      (f) Percentages of success frames on NYU

Figure 5. Comparison with the state-of-the-art methods on MSRA, ICVL, and NYU dataset. Top: The per-joint mean error for all the test examples. Bottom: Percentage of frames in the testing examples under different error thresholds.

Table 1. Comparison with the state-of-the-art methods.

| Methods | Mean Error [mm] ↓ | | | |
|---|---|---|---|---|
| | MSRA | NYU | ICVL | HANDS17 |
| HandPointNet [10] | 8.51 | 10.54 | 6.94 | - |
| DenseReg [29] | 7.23 | 10.21 | 7.30 | - |
| V2V-PoseNet [16] | 7.59 | 8.42 | 6.28 | 9.95 |
| Point-to-Point [11] | 7.71 | 8.99 | - | 9.82 |
| CrossInfoNet [5] | 7.86 | 10.08 | 6.73 | |
| A2J [30] | - | 8.61 | 6.46 | 8.57 |
| JGR-P2O [6] | 7.55 | 8.29 | 6.02 | - |
| HandFoldingNet [3] | 7.34 | 8.58 | 5.95 | - |
| SRN [20] | 7.16 | 7.78 | 6.26 | 8.39 |
| DenseRecurrent [4] | 7.01 | 6.85 | 6.05 | - |
| TriHorn-net [23] | 7.13 | 7.68 | 5.73 | - |
| IPNet [22] | - | 7.17 | - | - |
| HandR$^2$N$^2$ [2] | **6.54** | 7.27 | **5.71** | - |
| **DOR3D-Net (Ours)** | 6.93 | **6.71** | 5.87 | **6.99** |

Table 2. Comparison of heatmap-based methods on NYU dataset [28].

| Method | Backbone | Mean Error↓ [mm] |
|---|---|---|
| DOR3D-Net (w/ Heatmap Regression) | Resnet50 | 8.63 |
| **DOR3D-Net (w/ Ordinal Regression)** | **Resnet50** | **8.25** |

**Evaluation Metric.** We use two standard metrics to evaluate 3D hand pose estimation performance. The first one is the mean 3D Euclidean distance error (Mean Error) [30]. The second one is the percentage of success frames in which the worst joint 3D distance error is below a threshold [30]. Note that, the results of our method in this paper are all predicted by a single model.

## 4.2. Comparison with the State-of-the-art Methods

We compare our method with the state-of-the-art depth image-based hand pose estimation methods, i.e., Hand-PointNet [10], DenseReg [29], V2V-PoseNet [16], Point-to-Point [11], CrossInfoNet [5], A2J [30], JGR-P2O [6], HandFoldingNet [3], SRN [20], DenseRecurrent [4], TriHorn-net [23], IPNet [22], and HandR$^2$N$^2$ [2]. Fig. 5 shows the result of 3D per-joint mean error and the percentages of success frames over different error thresholds. Meanwhile, Tab. 1 shows the overall performance of DOR3D-Net and all the methods on four datasets, respectively. It can be seen that our method outperforms all the other methods on the NYU and HANDS17 datasets. On the ICVL dataset, our method is ranked third, with a slightly lower accuracy than the TriHorn-net [23] and HandR$^2$N$^2$ [2]. TriHorn-net [23] uses an innovative data agumentation approach and HandR$^2$N$^2$ [2] uses five iterative corrections. As noted in DenseRecurrent [4] and IP-Net [22], some of the 3D joint annotations in MSRA dataset contained significant errors. Therefore, the evaluation on the MSRA dataset may be less meaningful.

Table 3. Effectiveness of dense ordinal regression module.

| Method | Mean Error [mm] ↓ | | | |
|---|---|---|---|---|
| | x | y | z | all |
| DOR3D-Net (w/ offset-based regression) | 3.31 | 3.33 | 4.32 | 7.34 |
| **DOR3D-Net (w/ ordinal-based regression)** | **3.12** | **3.19** | **4.13** | **6.99** |

Table 4. Effectiveness of normal discretization.

| Method | Mean Error [mm] ↓ | | | |
|---|---|---|---|---|
| | x | y | z | all |
| DOR3D-Net (w/ uniform distribution) | 3.14 | 3.19 | 4.18 | 7.06 |
| **DOR3D-Net (w/ normal distribution)** | **3.12** | **3.19** | **4.13** | **6.99** |

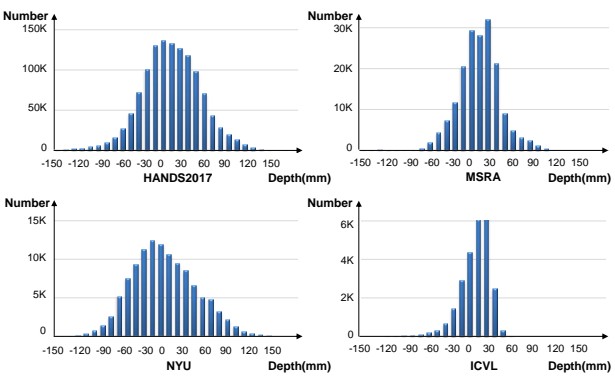

Figure 6. The depth distribution of joints on HANDS2017, MSRA, NYU, and ICVL, which are close to normal distribution.

Moreover, to further valid the effectiveness of the dense ordinal regression module, we compare it with the heatmap-based method [25] with the same backbone Resnet50 on NYU dataset [28]. Tab. 2 shows that our ordinal regression module surpasses the heatmap-based method. In a nutshell, DOR3D-Net shows significant superiority over other existing methods, indicating the benefit of the dense ordinal regression module.

## 4.3. Ablation Study

To demonstrate the effectiveness of each module in our method, we conduct extensive ablation studies on the HANDS2017 dataset. Tab. 3, Tab. 4, Tab. 5, Tab. 6 and Fig. 6 show the experimental results in detail.

**Effectiveness of dense ordinal regression module.** To verify the effectiveness of the dense ordinal regression module, we replace the regression module with an offset-based regression module and show the results in the Tab. 3. Specifically, we design the following model variants: (1) DOR3D-Net (w/ offset-based regression) : we train the model with the offset-based regression module; (2) DOR3D-Net (w/ ordinal-based regression) : we train the model with the dense ordinal regression module; For a fair comparison, we uses the same backbone and experimental

Table 5. Effectiveness of feature extractor module.

| Method | Module | Mean Error↓ [mm] | Params↓ [MB] | Speed↑ [FPS] |
|---|---|---|---|---|
| Input | DOR3D-Net (w/o UVMap) | 7.10 | **86.9** | **50** |
| | **DOR3D-Net(w/ UVMap)** | **6.99** | 86.9 | 47 |
| Backbone | DOR3D-Net (Resnet50-based) | 7.67 | **32.7** | **110** |
| | **DOR3D-Net (Transformer-based)** | **6.99** | 86.9 | 47 |
| Design | DOR3D-Net (w/ $F_{xyz}$ ) | 7.04 | **86.9** | 47 |
| | **DOR3D-Net (w/ $F_{xy}$&$F_z$)** | **6.99** | 86.9 | 47 |

configuration. As shown in Tab. 3, our ordinal-based regression method significantly outperforms the offset-based regression method, and the 3D mean error is reduced by 4.77% on HANDS2017 dataset. There are two main reasons: (1) Offset-based regression module regresses the hand joints in a large 3D solution space, which is hard to obtain the optimal solution. (2) The dense ordinal regression module predicts probability maps that vary smoothly with ordinal constraints and are insensitive to noise and outliers.

**Effectiveness of normal discretization.** To verify the effectiveness of normal discretizatione, we verified two discretization strategies to quantize the z-axis interval and show the results in Tab. 4. Specifically, DOR3D-Net (w/ uniform distribution) uses uniform distribution strategy. Similarly, DOR3D-Net (w/ normal distribution) uses normal distribution strategy. The results of DOR3D-Net (w/ uniform distribution) is worse than DOR3D-Net (w/ normal distribution), demonstrating verifying the effectiveness of normal distribution strategy. Moreover, we analyze statistics of the hand $z$ coordinate distribution in public four datasets (Fig. 6) and notice that it is close to normal distribution (ND).

**Effectiveness of feature extractor module.** In Sec. III-A, we proposes transformer-based feature extractor with three designs: (1) UVMap: since the transformer structure contains only relative positional embedding, we design to include UVMap in the input to provide global absolute spatial information; (2) Introduce transformer structure: it has the powerful capability to learn the long-range relationship of dense features; (3) Output feature design: considering the in-plane $xy$ regression and depth-plane $z$ regression are quite different, we output two features $F_{xy}$ and $F_z$ from transformer structure to regress the $xy$ and $z$ coordinates, respectively. We conduct an ablation study on the components in feature extractor and summarize our results in Tab. 5. The ablation study results show that each of designs in transformer-based feature extractor provides a meaningful contribution to improving model performance.

**Effectiveness of DOR Loss.** In this ablation, we investigate the effectiveness of dense ordinal regression loss (DOR loss) in our method. We design the following model variants: (1) DOR3D-Net (w/o DOR loss) : we train the model without the dense ordinal regression loss; (2) DOR3D-Net (w/o DOR loss) : we train the model with the dense ordinal regression loss; As can be seen from the Tab. 6, without

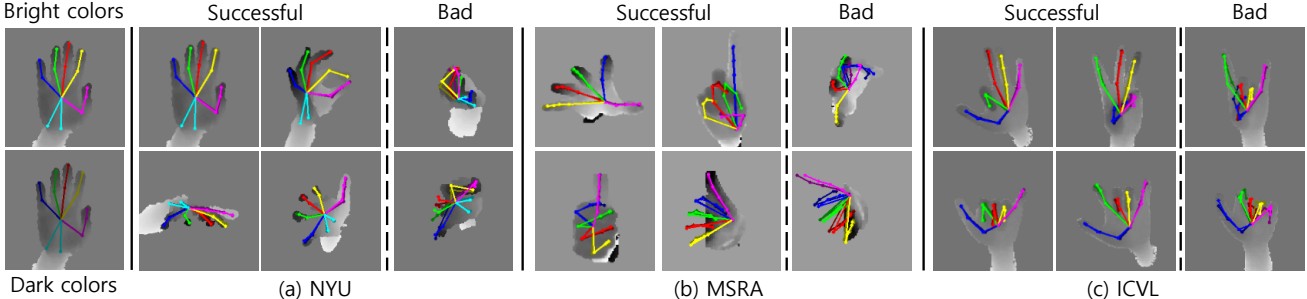

Bright colors    Successful      Bad      Successful      Bad      Successful      Bad

Dark colors        (a) NYU          (b) MSRA          (c) ICVL

Figure 7. Joints prediction on NYU, MSRA, and ICVL. Bright colors represent predicted results and dark colors show ground truth. The visualization results superimpose the predicted results on the ground truth. Both successful and bad cases are displayed.

Table 6. Effectiveness of DOR Loss. 'DOR loss' refers to the dense ordinal regression loss.

| Method | Module | Mean Error↓ [mm] | Params↓ [MB] | Speed↑ [FPS] |
|--------|--------|------------------|--------------|--------------|
| Loss | DOR3D-Net (w/o DOR loss) | 7.31 | **86.9** | **47** |
| | **DOR3D-Net (w/ DOR loss)** | **6.99** | 86.9 | 47 |

DOR loss, the error increased by 0.32mm. This also validates that dense probability supervision plays an important role in our DOR3D-Net to learn representative features and improve hand pose accuracy.

**Speed and Parameters.** We test our model on a single NVIDIA V100 GPU. The speed of the Resnet50-based backbone and transformer-based backbone is 110 FPS and 47 FPS, respectively. The full model meets the real-time requirement for practical applications. The parameters of Resnet50-based backbone and Transformer-based backbone are 32.7MB and 86.9MB, respectively.

**Qualitative Results.** Fig. 7 visualizes the successful and failure cases of our full model performance on MSRA dataset [26], NYU dataset [28] and ICVL dataset [27]. It can be seen that our method predicts well in most cases, while fails in cases of severe occlusion, large areas of missing pixels, and challenging viewpoint.

Moreover, in order to intuitively verify our method, we make some changes to the full model: (1) Replace the transformer with Resnet50; (2) Replace the ordinal regression with offset-based regression. Fig. 8 visualizes and compares the hand pose estimation results. In each image, the color in yellow, red, green, blue, and pink represents five fingers respectively. The bright colors denote estimated joints and the dark are the truth. It can be seen that our full model has better performance on edge blur, background noise and occlusion cases, while the other two variants predict worse joints' position which may suffer from outliers and low-level feature representation.

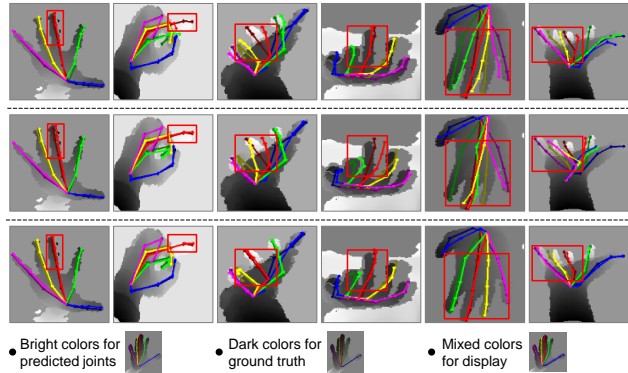

• Bright colors for predicted joints     • Dark colors for ground truth     • Mixed colors for display

Figure 8. Qualitative comparison results on HANDS2017. Top: DOR3D-Net (Resnet50-based). Middle: DOR3D-Net (w/ offset-based regression). Bottom: DOR3D-Net (full).

## 5. Conclusion

In this paper, we propose a DOR3D network that reformulates the 3D hand pose estimation as a dense ordinal regression problem. In comparison with offset-based regression methods, this formulation simplifies the solution space from a large interval to binary values which enables the network to learn easily and find out the optimal solution. Furthermore, a transformer-based feature extractor is utilized to enhance dense feature presentation and additional UV coordination maps are generated to provide absolute spatial information. Our DOR3D-net has achieved the SOTA performance on the HANDS2017, MSRA, NYU and ICVL datasets. This method provides robots and autonomous vehicles with the ability to accurately perceive human hand movements. Therefore, this technology is crucial for social-aware intelligent machines to function effectively in human-populated environments. In the future, we will improve the depth-based 3D hand pose estimation by improving the samples with the challenging scenes, such as extreme occlusions or very fast hand movements. Moreover, we will to improve the model efficiency for porting to robots or autonomous driving platforms for low-latency user interaction.

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
