# OpenReview forum: "DOR3D-Net: Dense Ordinal Regression Network for 3D Hand Pose Estimation"
_thecvf.com/CVPR/2024/Workshop/POETS — CVPR 2024 Workshop POETS Poster_

### Official Review · Reviewer_ZV3r · 2024-05-08
**This paper proposed a novel dense ordinary regression network DOR3D-Net to solve the 3D hand pose estimation. Extensive experiment results are provided to demonstrate the effectiveness and superiority.**

**Rating:** 6
**Confidence:** 5

**Review:**

Pros:
1. The paper is well-written and easy to follow.

2. The paper proposes an approach that solves the hand posed estimation task by a novel dense ordinary regression. Moreover, the new discretization method and ordinary regression loss are introduced.

3. Extensive experiments were conducted and demonstrate improvements.

Cons:
1. Missing recent SOTA literature. Table 1 should include the following recent published results for comparison.

[R1] Cheng W, Ko J H. HandR2N2: Iterative 3D Hand Pose Estimation Using a Residual Recurrent Neural Network[C]//Proceedings of the IEEE/CVF International Conference on Computer Vision. 2023: 20904-20913.

[R2] Ren P, Chen Y, Hao J, et al. Two heads are better than one: image-point cloud network for depth-based 3D hand pose estimation[C]//Proceedings of the AAAI Conference on Artificial Intelligence. 2023, 37(2): 2163-2171.

2. The best values in Table 5 should be highlighted as bold.

3. Line-322, how to obtain the "hand center point" for each dataset should be clarified. Calculating average location of GT joints? Or using the estimated center point provided by V2V [16]?

Minor editorial issue:

--Line-081, traing -> training

---

### Official Review · Reviewer_7z6Y · 2024-05-08
**Review of <DOR3D-Net: Dense Ordinal Regression Network for 3D Hand Pose Estimation>**

**Rating:** 6
**Confidence:** 2

**Review:**

**Abstract and Introduction**

The paper introduces DOR3D-Net, a novel architecture focusing on 3D hand pose estimation using a combination of dense ordinal regression and joint regression loss. The premise is based on overcoming the high computational costs and inaccuracies associated with earlier dense regression methods by utilizing binary classifiers for depth estimation.

**Technical Soundness and Innovation**

The technical framework is solid, leveraging the strengths of transformers for feature extraction and introducing a novel DOR module for 3D localization. This dual approach is innovative in its handling of depth estimation, addressing the significant challenge of precision in 3D pose estimation tasks. However, as noted below, the idea of using ordinal regression is not entirely novel, echoing techniques used in full-body pose estimation tasks as indicated by references to similar works.

**Critical Observations**

- Although the results are promising, the paper would benefit from a deeper exploration into the failures or limitations of the proposed method, especially under challenging conditions such as extreme occlusions or very rapid hand movements.
- The computational efficiency, although addressed, might still be a concern for real-time applications, as transformers are generally more computationally intensive. Exact comparisons in terms of inference time against other methods under similar hardware conditions would add more depth.

**Pros and Cons**
- **Pros:**
  - The innovative combination of joint and ordinal regression losses enhances model performance and robustness.
  - The method achieves state-of-the-art results, which are well-supported by extensive experiments and comparisons.

- **Cons:**
  - The novelty is somewhat diluted by the similarity in approach to existing methods in related fields, which could be a point of contention regarding the claimed contributions.
  - The paper does not address the limitations related to occlusions thoroughly, which are critical in real-world applications of hand pose estimation.

**Discussion and Questions**

The discussion around the pipeline's close resemblance to the original Swin Transformer and the modifications in stage-3 could have been more detailed, offering deeper insights into the choices that led to these design decisions. The absence of a follow-up on handling occlusions, as noted in the review, also leaves a gap in the discussion that could be crucial for practical applications.

It would be beneficial to test the model in real-world scenarios, possibly integrating it with real-time video input to evaluate performance dynamically. Expanding the discussion on the model's limitations by analyzing more thoroughly the conditions under which it fails could guide future improvements.

**Conclusion**

Overall, the paper presents a well-thought-out approach to tackling the problem of 3D hand pose estimation, introducing a robust method to handle the complexities associated with depth data. Despite the challenges in novelty and handling occlusions, the results are compelling, though some areas may require further exploration and validation in real-world conditions.

---

### Official Review · Reviewer_Uw9D · 2024-05-11
**Acceptable paper - suitability for POETS workshop uncertain**

**Rating:** 4
**Confidence:** 3

**Review:**

This paper addresses the task of hand pose estimation from depth imagery. The authors reformulate the 3D pose estimation problem as a dense ordinal regression problem and introduce Dense Ordinal Regression 3D Pose Network (DOR3D-Net) for 3D hand pose estimation. The DOR3D-Net simplifies the solution space to binary values, reducing noise and improving learning efficiency. It uses a transformer-based feature extractor and UV coordination maps for enhanced spatial information. The network has achieved state-of-the-art performance on the HANDS2017, MSRA, NYU, and ICVL datasets.

Pros:
- Improve upon previous state-of-the-art methods (Sec. 4.2)
- Thorough ablation (Sec 4.3), motivating the design choices
- Visualization of both successful and unsuccessful predictions

Cons:
- The improvements are minor and could be considered within noise. Considering this, it would've been more appropriate to report an average of multiple seeded runs to make sure the improvements are not within error. This is exacerbated by the fact that the annotations of NYU/ICVL are noisy.
- The suitability of this paper in the POETS workshop is debatable. This paper address 3D hand pose estimation from depth imagery using standard benchmarking datasets. The POETS workshop webpage states that the workshop is about virtual humans for robotics and autonomous driving. None of these topics are addressed. Therefore I recommend resubmitting this paper to a pose workshop.

---

### Meta-Review · Program_Chairs · 2024-05-14

**Recommendation:** Accept (Poster)
**Confidence:** 5

**Metareview:**

This paper introduces the DOR3D-Net for 3D hand pose estimation using dense ordinal regression, simplifying the solution space and enhancing learning efficiency. While achieving state-of-the-art results, the paper faces criticism for minor improvements and relevance to the workshop theme. Despite its shortcomings, the solid technical framework and extensive experimental validation support its acceptance, with recommendations for addressing noted gaps in real-world application and novelty. However, the discussion of virtual humans in autonomous driving and robotics should be covered in the final version to match the workshop topic in the final version.

---

### Decision · Program_Chairs · 2024-05-14

Accept (Poster)